# Seabed video and still images from the northern Weddell Sea and the western flanks of the Powell Basin

Autun Purser[1], Simon Dreutter[1], Huw Griffiths[2], Laura Hehemann[1], Kerstin Jerosch[1], Axel Nordhausen[3], Dieter Piepenburg[1,4], Claudio Richter[1], Henning Schröder[1], Boris Dorschel[1]

[1] Alfred Wegener Institute, Helmholtz Centre for Polar and Marine Research Bremerhaven, Am Handelshafen 12, 26570 Bremerhaven, Germany

[2] British Antarctic Survey, High Cross, Madingley Rd, Cambridge, CB3 OET, UK

[3] Max Planck Institute for Marine Microbiology, Celsiusstrasse 1, 28359, Bremen, Germany

[4] Helmholtz Institute for Functional Marine Biodiversity at the University of Oldenburg (HIFMB), Carl-von-Ossietzky-Str. 9–11, 26129 Oldenburg, Germany

*Correspondence to*: Autun Purser (autun.purser@awi.de)

**Abstract.** Research vessels equipped with fibreoptic and copper cored coaxial cables support the live onboard inspection of high-bandwidth marine data in real-time. This allows towed still image and video sleds to be equipped with latest generation higher resolution digital camera systems and additional sensors. During RV *Polarstern* expedition PS118 in February-April 2019, the recently developed Ocean Floor Observation and Bathymetry System (OFOBS) of the Alfred Wegener Institute was used to collect still and video image data from the seafloor at a total of 11 predominantly ice covered locations in the northern Weddell Sea and Powell Basin. Still images of 26 megapixel resolution and HD quality video data were recorded throughout each deployment. In addition to downward facing video and still image cameras, OFOBS also mounted sidescan and forward-facing acoustic systems, which facilitated safe deployment in areas of high topographic complexity, such as above the steep flanks of the Powell Basin and the rapidly shallowing, iceberg scoured Nachtigaller Shoal. To localise collected data, the OFOBS system was equipped with a POSIDONIA transponder for Ultra Short Baseline triangulation of OFOBS positions. All images are available from: https://doi.org/10.1594/PANGAEA.911904 (Purser et al., 2020).

## 1 Introduction

Recent studies indicate that climate change processes in the maritime Antarctic are accelerating, leading to the increasing retreat of the ice sheets of the eastern coast of the Antarctic Peninsula and the near disintegration of the ice shelves Larsen A in 1995 and Larsen B in 2002 (Rott et al., 1996; Scambos et al., 2013; Shepherd et al., 2003), as well as the break-off of giant icebergs from Larsen C (Han et al., 2019; Shepherd et al., 2003; Skvarca, 1993). This change in ice coverage has resulted in spatially extensive habitat change both above and below water, with many processes from light penetration, ocean stratification, surface productivity and food transportation pathways affected (Barnes and Tarling, 2017; Chaabani et al.,

2019; Fillinger et al., 2013; Griffiths et al., 2017; Gutt et al., 2019). By assessing the seafloor habitats and associated benthic fauna communities in these areas recently free of permanent ice cover with the Ocean Floor Observation and Bathymetry

System (OFOBS) of the Alfred Wegener Institute (AWI), Helmholtz Centre for Polar and Marine Research Bremerhaven (Germany) (**Figure 1**), the ecological effects of climate-driven ice shelf loss can be studied. OFOBS is a towed platform capable of deployment in moderate ice-cover conditions and capable of concurrently collecting acoustic, video and still image data from the seafloor (Purser et al., 2018).

During RV *Polarstern* expedition PS118 (February 9$^{th}$ – April 10$^{th}$ 2019) (Dorschel, 2019b) OFOBS was deployed 11 times, conducting concurrent high-resolution still image, video and acoustic surveys across diverse and contrasting regions of the Antarctic seafloor (**Figure 2**). Ice conditions during the cruise were harsh, though a north-south transect from the Weddell Sea continental shelf to the northwest continental shelf edge of the Powell Basin was completed, with stations on the Weddell Sea plateau, Nachtigaller Shoal (Dorschel et al., 2014) and the flanks and rim of the Powell Basin investigated

(**Table 1**). The forward facing acoustic camera allows the towed OFOBS operator to be aware of steep structures or rising seafloor ahead of the device, and allow ample time to winch the system to a safer height. This allowed use even in heavy ice, with minimal ability for ship manoeuvre, and close to the steep flanks of the Powell Basin, allowing collection of this novel dataset.

The PS118 OFOBS data was collected with the same cameras, illumination regime and deployment protocols as mounted on the previous AWI towed camera sled system, (Ocean Floor Observation System (OFOS)), used to survey the Antarctic seafloor during previous recent RV *Polarstern* expeditions (Piepenburg et al., 2017), such as PS81, also to the western Weddell Sea area in 2013 (Gutt, 2013; Gutt et al., 2016) and PS96, to the southeastern Weddell Sea in 2015/16 (Schröder et al., 2016). By continuing to mount the same camera systems, observations made during PS118 can be most readily compared

with those made during the previous expeditions, uncomplicated by methodological problems relating to variabilities in camera performance, flight height or illumination (Schoening et al., 2020).

The OFOBS data collected during PS118 and presented here is of use for a range of scientific studies, such as:

1) Assessment of epibenthic megafauna communities observed at several sites along a South-North transect from the
Weddell Sea Antarctic Peninsula continental shelf to the Powell Basin.

2) For spatial comparison of these observed shelf and basin megafauna communities with those observed with the very similar OFOS towed device sled during recent cruises.

3) A temporal picture of seafloor communities occupying these seafloor regions in 2019, for comparison with future studies following continued ice loss.

4) An extensive set of seafloor images from several locations on the Powell Basin flank, a region of southern seafloor sparsely surveyed to date.

## 2 Materials and methods

### 2.1 Ocean Floor Observation and Bathymetry System (OFOBS)

The OFOBS is a state-of-the-art towed camera and acoustic survey sled recently developed by the Deep Sea Ecology and Technology group of AWI for benthic polar research in ice-covered environments (Purser et al., 2018). The device was deployed during PS118 as described in Purser et al. (2018), taking images under comparable illumination conditions, flight heights and with the same camera systems as were mounted on the OFOS sled during the PS86 and PS96 cruises in 2013 and 2015/16, respectively (Piepenburg et al., 2017). OFOBS positioning during deployments was carried out with the iXBlue POSIDONIA Ultra-Short Base Line (USBL) system used by RV *Polarstern*, localising the relative position of the OFOBS to the vessel (itself deriving its position from a satellite based Global Navigation Satellite System (GNSS). Every few seconds (depending on deployment depth) the OFOBS received a new position fix, which was used to position stamp each collected image against a UTC timestamp. During PS118, a stable position fix was attained with an accuracy of approximately 0.2% of the slant range from the ship to the subsea unit. Environmental and operational factors, such as ice coverage and vessel speed could result in a slanted tether cable, though for the deployment depths made during PS118 a positional accuracy of within ~20 m was likely maintained throughout. Images were taken at 26 megapixels resolution with the camera system (iSiTEC, CANON EOS 5D Mark III) with an automated timer (for the majority of images, these were taken with a frequency of ~1 image every 20 seconds) Additional images taken at the discretion of the operator. These two image categories are distinguished in the data set as 'HOTKEY' and 'TIMER' images – designations automatically incorporated into the timestamped filenames. 'HOTKEY' images were commonly taken on the first observation of a particular fauna species or to record a feature of interest, such as a whale fall or interesting geological structure. No processing stages were applied to the collected data, with the native camera .jpeg data provided in the dataset at full acquired resolution.

Throughout all of these deployments the OFOS and OFOBS systems were equipped with three red sizing lazers (FLEXPOINT) (Purser et al., 2018; Purser and Sablotny, 2020), arrayed in an equilateral triangle with 50 cm spacing around the still camera housing. This lazer array ensures that each image recorded has three red dots near the centre of the image, each spaced by 50 cm. These dots can be used in subsequent analysis to determine accurately the area covered by a particular image. Throughout PS118 the OFOBS was deployed approximately 1.5 – 2m above the seafloor, giving a coverage within each collected image of $4 – 6$ m$^2$.Illumination of the seafloor was provided by four downward facing SeaLight sphere 3150 LED lights positioned in the corners of the main OFOBS frame, with two additional strobe lights (iSiTEC UW-Blitz 250,

TTL driven) firing concurrently with image collection.Throughout all deployments, HD video data was recorded by the OFOBS for the duration of each dive with an HD video recorder (iSiTEC, Sony FCB-H11) (Purser et al., 2018).

## 2.2 Field sampling

### 2.2.1 Weddell Sea sampling

OFOBS surveys were carried out in a roughly south-north transect from the Weddell Sea continental shelf of the Antarctic Peninsula to the northern Powell Basin. In total, 7 OFOBS deployments were made between 65 and 62$^o$S (**Table 1**, **Figure 2**). With the exception of an OFOBS deployment across the Nachtigaller Shoal (Dorschel et al., 2014) (station PS118/11-2), the seafloor in these regions was observed to be predominantly made up of soft material with occasional drop stones present (**Figure 3**).


### 2.2.2 Powell Basin sampling

Four successful OFOBS deployments were made on the flanks and rim of the Powell Basin (**Table1, figure 2**). Three of these deployments, PS118/39-1, PS118/69-1 and PS118/81-1, benefitted from the equipping of the OFOBS sled with a forward looking sonar (Purser et al., 2018). This sensor allowed OFOBS to be used over very steep terrain with minimum
risk, by giving advance warning of any hard structures ~30 m ahead of the sled. The majority of towed sleds, such as OFOS, are less capable in high-relief regions, where snagging on tall structures such as cliffs or vent structures can occur. With OFOBS, the ~30 m warning of approach is sufficient to allow the operators to commence winching of the system in good time to minimise the risk of impact of the sled with the seafloor whilst still collecting usable image data. These data sets each cover many 100s of meters of the Powell Basin flank walls, visually surveying these traditionally difficult to investigate
regions of seafloor (**Figure 4**).

## 3 Data availability

All seafloor images collected with the OFOBS system are available from the data publisher PANGAEA. No preprocessing or processing stages were applied prior to upload, with no colour correction or light vignetteing algorithms applied. These
images are provided with georeferenced positions for each image, as derived from the POSIDONIA system (https://doi.org/10.1594/PANGAEA.911904; (Purser et al., 2020)). The full cruise track is also available via PANGAEA (https://doi.org/10.1594/PANGAEA.901319; (Dorschel, 2019a)) with information on additional environmental and scientific data collected during the cruise given in the PS118 cruise report (Dorschel, 2019b). Video data collected via the OFOBS system is available from the authors on request. In addition to the image data presented in this paper, the multibeam data

concurrently collected by the OFOBS device has also been uploaded to PANGAEA and will be available from April 2021 for open access download, or on request from the authors.

**Competing interests.**

The authors declare that they have no conflict of interest.

**Author contributions.**

AP applied for the secondary user time for the PS118 cruise, conceived of the investigation and ran the data collection campaign. BD was chief scientist for the PS118 expedition. SD, LH, HS, HG and AN helped run the OFOBS platform. AP, HG, KJ, DP, CR and BD determined sampling strategies for the OFOBS and aided in data collection. AP prepared the 140 manuscript with contributions from all co-authors.

**Acknowledgements.**

The captain and crew of RV *Polarstern* expedition PS118 are thanked for their support and interest in the OFOBS deployments conducted during the cruise. U Hoge is thanked for his assistance in installing the OFOBS system prior to cruise commencement.

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

**FIGURES**

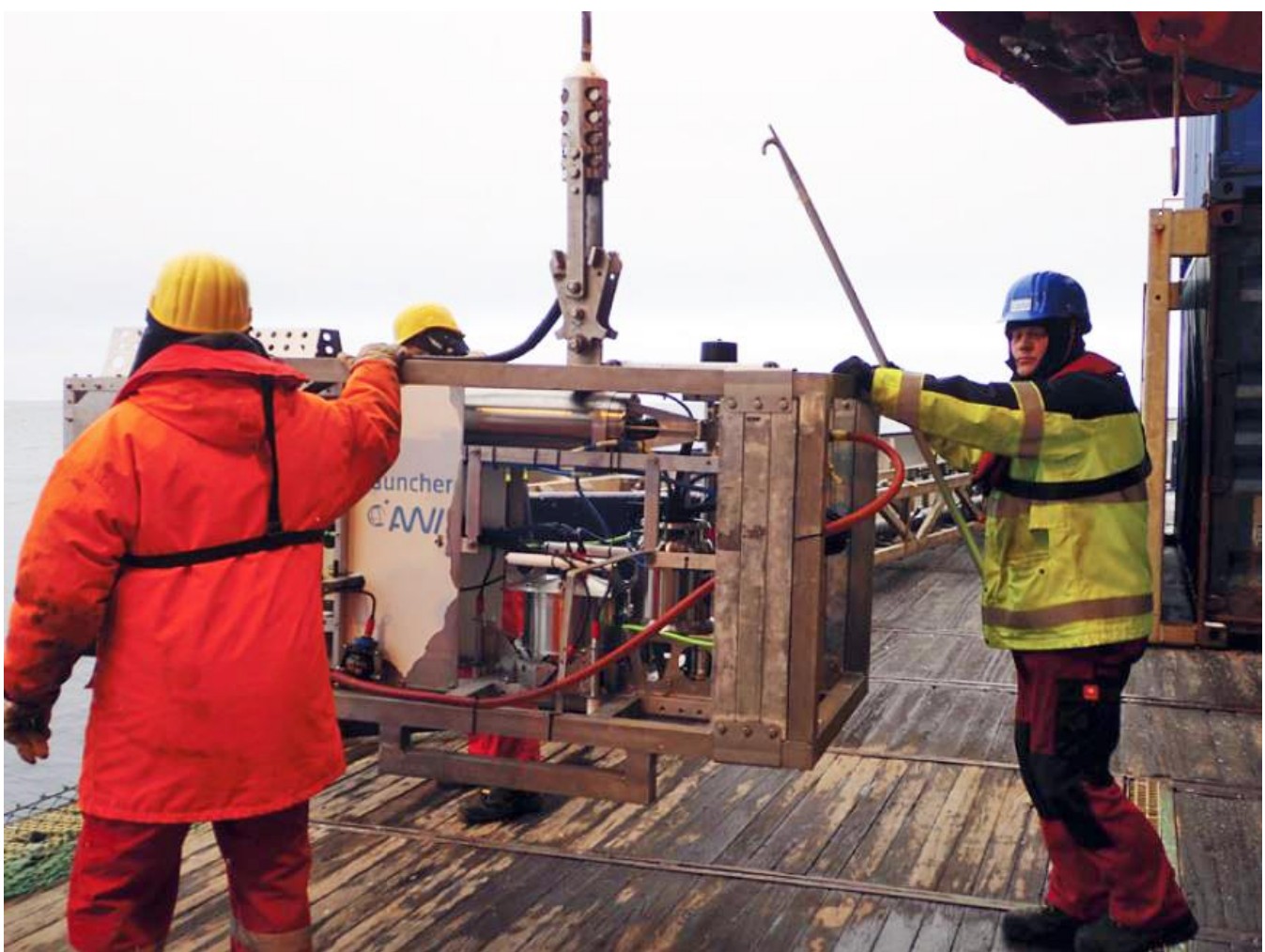

**Figure 1:** The Ocean Floor Observation and Bathymetry System (OFOBS) of the Alfred Wegener Institute (AWI), Helmholtz Centre for Polar and Marine Research Bremerhaven, deployed from the RV *Polarstern* during cruise PS118 in the waters east of the Antarctic Peninsula and on the flanks of the Powell Basin (Feb – April 2019)


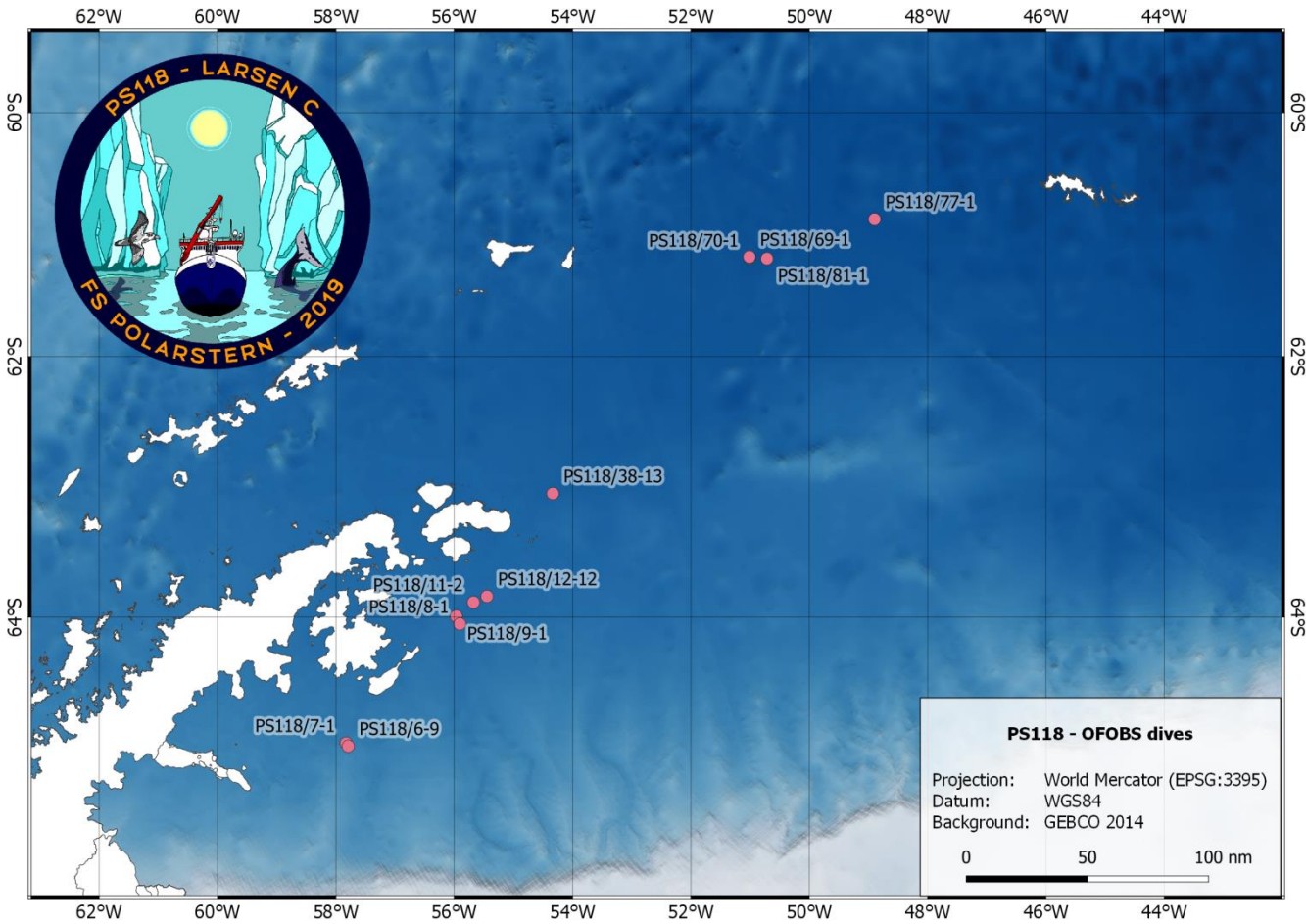

**Figure 2:** Regional map showing the positions of OFOBS deployments made during RV *Polarstern* cruise PS118.


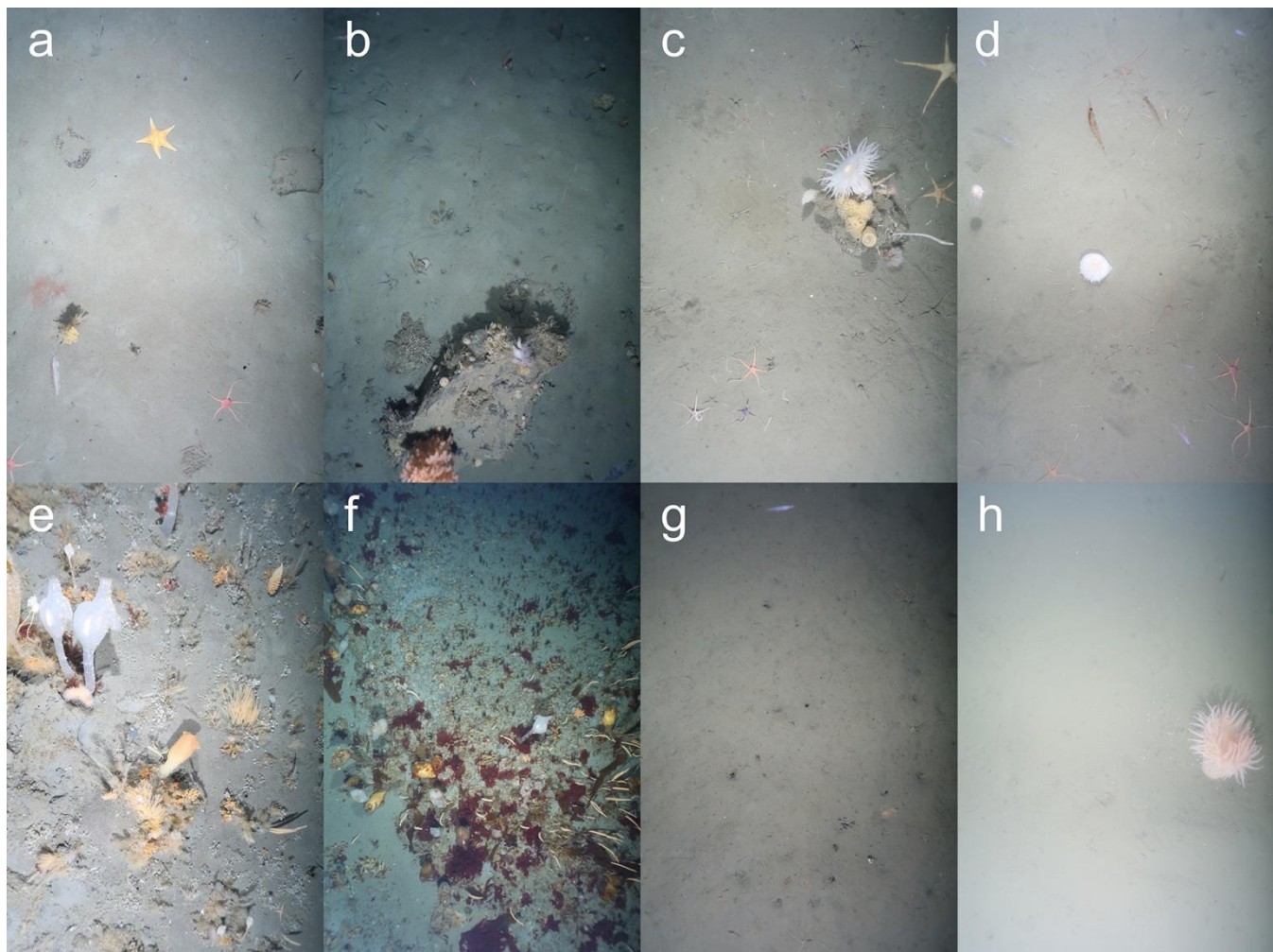

**Figure 3:** Typical seafloor images collected from each of the OFOBS surveys made of the Weddell Sea seafloor during RV *Polarstern* cruise PS118. a) Station PS118_6, hotkey_2019_03_06 at 05_07_31 b) Station PS118_7,  hotkey_2019_03_06 at 19_36_28 c) Station PS118_8, hotkey_2019_03_11 at 14_33_24   d) Station PS118_9, timer_2019_03_12 at 06_01_06   e) Station PS118_11, timer_2019_03_13 at 20_14_04   f) Station PS118_11,  timer_2019_03_13 at 22_18_47  g) Station PS118_12, timer_2019_03_15 at 01_29_36 h) Station PS118_38, hotkey_2019_03_23 at 06_27_00. All images are presented here and in the dataset with no manipulation or colour correction.

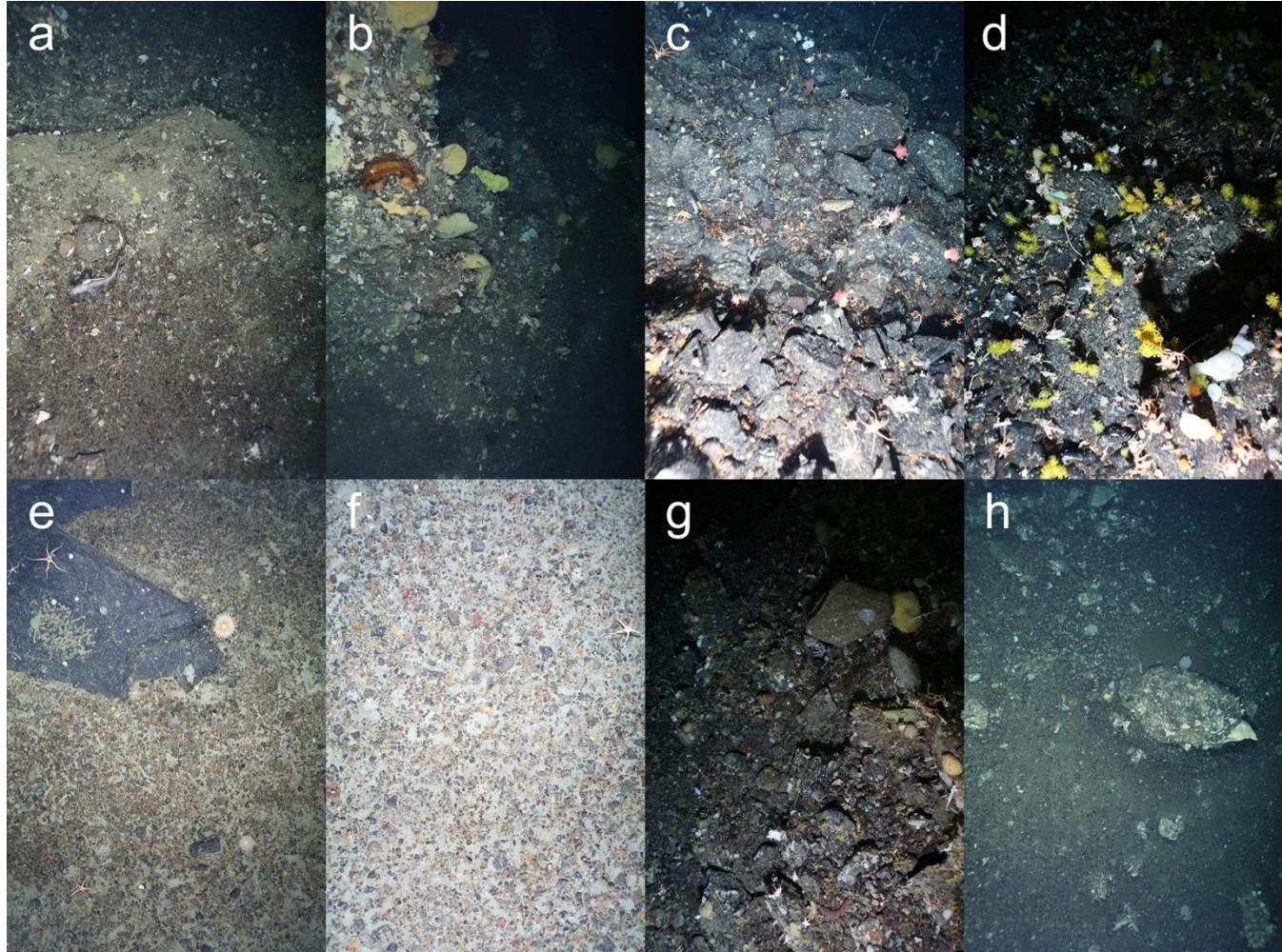


**Figure 4:** Typical seafloor and cliff escarpment images collected during OFOBS surveys of the Powell Basin flanks during RV *Polarstern* cruise PS118 . a) Station PS118_39, hotkey_2019_03_23 at 23_21_46 b) Station PS118_39, timer_2019_03_25 at 18_50_09 c) Station PS118_69, hotkey_2019_03_31 at 12_20_55 d) Station PS118_69,
hotkey_2019_03_31 at 14_10_56   e) Station PS118_77, timer_2019_04_01 at 20_39_51   f) Station PS118_77, timer_2019_04_01 at 20_11_32  g) Station PS118_81, timer_2019_04_04 at 00_25_31 h) Station PS118_81, timer_2019_04_04 at 02_27_33. All images are presented here and in the dataset with no manipulation or colour correction.

**TABLES**

**Table 1:** Locations of OFOBS deployments during PS118. Start and end position co-ordinates and times are given, in addition to the numbers of 'timer' and 'hotkey' images collected during each deployment.

| Station Number | Region | Date | Start (UTC) | End (UTC) | Latitude Start | Longitude Start | Latitude End | Longitude End | Timer Images | Hotkey Images |
|---|---|---|---|---|---|---|---|---|---|---|
| PS118_6 | Weddell Sea | 06.03.2019 | 05:01 | 09:33 | -64.939363 | -57.791526 | -64.929378 | -57.818072 | 1314 | 90 |
| PS118_7 | Weddell Sea | 06.03.2019 | 19:25 | 20:08 | -64.918268 | -57.824799 | -64.914652 | -57.826477 | 259 | 28 |
| PS118_8 | Weddell Sea | 11.03.2019 | 12:02 | 15:45 | -64.049462 | -55.906690 | -64.007332 | -55.908186 | 891 | 73 |
| PS118_9 | Weddell Sea | 12.03.2019 | 06:00 | 06:51 | -63.991755 | -55.965738 | -63.993996 | -55.968168 | 319 | 4 |
| PS118_11 | Nachtigaller | 13.03.2019 | 20:14 | 22:25 | -63.888204 | -55.675914 | -63.887744 | -55.656926 | 690 | 7 |
| PS118_12 | Weddell Sea | 15.03.2019 | 01:28 | 05:55 | -63.844375 | -55.447947 | -63.844602 | -55.446939 | 1547 | 5 |
| PS118_38 | Weddell Sea | 23.03.2019 | 05:01 | 06:31 | -63-066654 | -54.334116 | -63.073675 | -54.319400 | 379 | 11 |
| PS118_39 | Powell Basin | 23.03.2019 | 18:46 | 09:56 | -61.914657 | -53.323264 | -61.865707 | -53.324424 | 3667 | 140 |
| PS118_69 | Powell Basin | 31.09.2019 | 01:52 | 16:00 | -61.192316 | -50.996506 | -61.192854 | -51.085361 | 2655 | 68 |
| PS118_77 | Powell Basin | 01.04.2019 | 21:08 | 21:48 | -60.888864 | -48.896518 | -60.887241 | -48.905883 | 153 | 5 |
| PS118_81 | Powell Basin | 03.04.2019 | 21:24 | 09:31 | -61.213897 | -50.716745 | -61.186405 | -50.672902 | 373 | 12 |