# Peer review of "Seabed video and still images from the northern Weddell Sea and the western flanks of the Powell Basin"

_Earth System Science Data, 2020_

## Referee Comment (RC1) · Anonymous Referee #1 · 2 Nov 2020

This data manuscript outlines the results of a field campaign using a novel acoustic-assisted camera system in a region currently under-sampled by traditional methods due to permanent ice cover. This is a valuable contribution to the scientific community and the data provided could support a variety of analyses that the authors suggest. The data manuscript provides the metadata necessary for use of the image dataset as well as information about the camera system and data collection sites. The only information lacking (which may be present in other manuscripts referenced) is whether the images undergo any color-correction as these appear to be corrected in the panels of figures 3 and 4, and the height above the seafloor at which images are captured to determine the seafloor area captured per image. Though this information may be

documented elsewhere, it is important to include to ensure proper use of the data set by other researchers. These missing data are imperative for the accurate calculation of abundances, for example. Reporting of color-correction methods is standard for image data sets, and is important for future use for taxonomy, etc. The description of the camera sled itself should be explicitly referenced (i.e. "a full description of the OFOBS system and its components are available in ____.") and if it does not exist, then a description (even brief) should be added here. A short description with technical specs of the cameras and strobes is at least provided in the downloaded file - it should be clear this information exists prior to downloading by an interested user. This data manuscript is very short but provides the necessary information for use of the dataset in a succinct manner.

I recommend publishing this data manuscript with the following minor edits: 1) add information regarding color-correction or explicitly reference where this information/methodology can be found 2) explicitly reference the full description of the vehicle and its components. If this does not exist then a description should be added here 3) note the height above the seabed at which images were collected and thus the approximate seafloor area covered by each image 4) line 30 - this change in ice cover has not have 5) missing ) on line 78 after (GNSS)

---

## Author Comment (AC1) · 3 Nov 2020

Thank you for this valuable input on this data paper.

We will make sure we incorporate the following information into the revised draft:

1) No colour correction has been conducted. We prefer to upload unpreocessed data for these data sets, to allow users to process further as required.

2) The OFOBS vehicle is described in detail in the refererenced paper, but we will summarise that paper here in a general paragraph, and elaborate on the camera used in greater detail.

[Figure]

3) Our ideal flight height is 1.5 m above seafloor, but difficult to achieve in the high relief Powell Basin area. Our lazers give a size estimation for seafloor coverage. These points will be made clearer in the revised version.

4 - 5) These minor gramattical errors will be addressed.

I think these points will make the data more straightforward for analysis, so thanks again!

Autun (on behalf of all authors)

---

## Referee Comment (RC2) · Katrien Van Landeghem (Referee) · 6 Nov 2020

This short, but well-written manuscript describes two seafloor survey campaigns with the Ocean Floor Observation and Bathymetry System (OFOBS) and the significance of surveying the seabed in these areas. This camera sled with forward looking sound-waves is designed to safely operate in partly sea ice covered oceans and in areas where seafloor bathymetry varies quickly. Both these conditions have traditionally hampered seafloor studies of areas where the impact of the climate crisis on West Antarctic Peninsula seafloor habitats are happening at a fast pace, and therefore critical for us to better understand wider ecological effects (see also Barnes et al., 2020 – GCB 26,

2750-2755).

I recommend publishing this data manuscript, but I have a few minor recommendations to take into consideration to allow more users to interact with the dataset:

1. The acoustic element of the OFOBS is mentioned several times – both the forward looking acoustics and the integrated side scan sonar (SSS). The former is a truly great asset to safely survey seafloor with variable topography. The latter, the SSS, is not represented in the data images or downloadable datasets. Such high-resolution and high-quality data of backscatter intensity from just a few meters above the seabed is invaluable to assess the relationship between seafloor habitats and acoustic backscatter, and the impact of this data manuscript could be much higher if some acoustic backscatter data was visualised to capture this potential.

2. In relation to point 1 above, geo-positioning is a key component of potential time-lapse analyses from repeated surveys. Do the authors have a handle on the confidence intervals of the USBL positioning? This is a minor point out of interest, really, but I suspect the positioning is very good and once again show-cases the potential for the data to be used to study both spatial and temporal changes.

3. The video images are left unprocessed for people to download, that is good practice. For the purpose of the short manuscript, I would recommend that at least some example images are published in a processed form. That would allow the laser pointers to become visible (providing a scale, which they really need to have), and it would show-case the full potential of detail achievable with the OFOBS.

4. The authors don't emphasise enough in my opinion that the OFOBS allows seabed surveys in partly sea ice covered areas. In fact, "ice conditions were harsh" – L42, and the abstract could mention that accomplishment specifically I feel.

---

## Author Response (AR1)

**RE: essd-2020-233: Seabed video and still images from the northern Weddell Sea and the western flanks of the Powell Basin**

Dear ESSD editorial team and reviewers,

5

10

We would like to thank the ESSD team and the reviewers for taking the time to read our manuscript and put forward some sensible suggestions on how we can present the data in a more immediately useful way and give a better understanding of the strengths and drawbacks of the data. We have incorporated the suggestions into this revised and resubmitted version of the manuscript, and we present how we incorporated the various suggestions here below. A 'tracked

**Anonymous Referee #1**

Received and published: 2 November 2020

changes' version of the manuscript then follows.

- 15 This data manuscript outlines the results of a field campaign using a novel acoustic-assisted camera system in a region currently under-sampled by traditional methods due to permanent ice cover. This is a valuable contribution to the scientific community and the data provided could support a variety of analyses that the authors suggest. The data manuscript provides the metadata necessary for use of the image dataset as well as information about the camera system and data collection sites.
- 20 The only information lacking (which may be present in other manuscripts referenced) is whether the images undergo any colorcorrection as these appear to be corrected in the panels of figures 3 and 4, and the height above the seafloor at which images are captured to determine the seafloor area captured per image. Though this information may be documented elsewhere, it is important to include to ensure proper use of the data set by other researchers. These missing data are imperative for the accurate calculation of
- 25 abundances, for example. Reporting of colorcorrection methods is standard for image data sets, and is important for future use for taxonomy, etc. The description of the camera sled itself should be explicitly referenced (i.e. "a full description of the OFOBS system and its components are available in \_\_\_\_\_.") and if it does not exist, then a description (even brief) should be added here.

30 Thank you for this observation. As you suggest, much of this information is indeed in the referenced manuscripts, but for ease of reading, we place it directly into this revised version of the manuscript. All figures presented in the paper are actually not colour corrected, and presented as raw images, which holds true also for the uploaded PANGAEA data set. We agree that undocumented colour (and lens) corrections can cause problems for later data users, so we prefer to avoid that by uploading the most basic form of collected image as possible to the archive. This is now made clear in the text and figure heading.

A short description with technical specs of the cameras and strobes is at least provided in the downloaded file - it should be clear this information exists prior to downloading by an interested user. This data manuscript is very short but provides the necessary information for use of the dataset in a succinct manner.

40

We have made a more complete description of the OFOBS camera equipment in this revised version of the manuscript.

I recommend publishing this data manuscript with the following minor edits:

45

1)add information regarding color-correction or explicitly reference where this information/methodology can be found

Done, as described above.

2) explicitly reference the full description of the vehicle and its components. If this does not exist then a description should be added here
 Improved, as described above.

3)note the height above the seabed at which images were collected and thus the approximate seafloor area

55 covered by each image

The text relating to image coverage has been improved within this version of the manuscript.

4) line 30 - this change in ice cover has not have

Changed as suggested.

5) missing ) on line 78 after (GNSS). Corrected as suggested.

**65**

**Referee #2**

**k.v.landeghem@bangor.ac.uk**

Received and published: 6 November 2020

- This short, but well-written manuscript describes two seafloor survey campaigns with the
  Ocean Floor Observation and Bathymetry System (OFOBS) and the significance of surveying the seabed in these areas. This camera sled with forward looking sound-waves is designed to safely operate in partly sea ice covered oceans and in areas where seafloor bathymetry varies quickly. Both these conditions have traditionally hampered seafloor studies of areas where the impact of the climate crisis on West Antarctic Peninsula
  seafloor habitats are happening at a fast pace, and therefore critical for us to better understand wider ecological effects (see also Barnes et al., 2020 GCB 26, 2750-2755). I recommend publishing this data manuscript, but I have a few minor recommendations to take into consideration to allow more users to interact with the dataset:
- 1. The acoustic element of the OFOBS is mentioned several times both the forward looking acoustics and the integrated side scan sonar (SSS). The former is a truly great asset to safely survey seafloor with variable topography. The latter, the SSS, is not represented in the data images or downloadable datasets. Such high-resolution and high-quality data of backscatter intensity from just a few meters above the seabed is invaluable
- to assess the relationship between seafloor habitats and acoustic backscatter, and the impact of this data manuscript could be much higher if some acoustic backscatter data was visualised to capture this potential.

We agree with this comment on the usefulness of the acoustic data collected by the 90 OFOBS during PS118. We intended to publish processed seafloor maps generated from this data in the future, but we have decided to take this comment on board and upload the data to PANGAEA to further support the image dataset. This data will be available in April 2021 and this has been made clear in the revised manuscript. We maintain the focus of this revised manuscript on the image data, but point the interested reader in the right 95 direction to acquire the acoustic data.

2. In relation to point 1 above, geo-positioning is a key component of potential time-lapse analyses from repeated surveys. Do the authors have a handle on the confidence intervals of the USBL positioning? This is a minor point out of interest, really, but I suspect the positioning is very good and once again show-cases the potential for the data to be used to study both spatial and temporal changes.

100

The accuracy of this is now presented more clearly in the revised text (0.2% of slant angle resolution a rough estimate of accuracy).

105

110

115

3. The video images are left unprocessed for people to download, that is good practice. For the purpose of the short manuscript, I would recommend that at least some example images are published in a processed form. That would allow the laser pointers to become visible (providing a scale, which they really need to have), and it would showcase the full potential of detail achievable with the OFOBS.

As mentioned in response to the previous reviewer, the seafloor covered by each image is now better described in the text. Although the lazers are difficult to spot in the small images within figs 3 and 4 they are actually quite distinct in the majority of images within the dataset when downloaded and viewed.

4. The authors don't emphasise enough in my opinion that the OFOBS allows seabed surveys in partly sea ice covered areas. In fact, "ice conditions were harsh" – L42, and the abstract could mention that accomplishment specifically I feel.

120 This aspect of the OFOBS was described in one of the cited papers, but we have highlighted this usefulness in the revised version of the manuscript.

Thank you for taking the time to make these suggestions for our manuscript!

125

On behalf of all authors, yours sincerely,

Autun Purser

**TRACK CHANGES VERSION OVERLEAF:**

**Seabed video and still images from the northern Weddell Sea and the western flanks of the Powell Basin**

Autun Purser1, Simon Dreutter1, Huw Griffiths2, Laura Hehemann1, Kerstin Jerosch1, Axel Nordhausen3, Dieter Piepenburg1,4, Claudio Richter1, Henning Schröder1, Boris Dorschel1

135

[revised manuscript text omitted]
 <a href="http://doi.org/10.1109/JOE.2018.2794095">http://doi.org/10.1109/JOE.2018.2794095</a> , hdl:10013/enic.da1678ed b2c3 4015-

375 - doi:10.1109/JOE.2018.2/94095 <http://doi.org/10.1109/JOE.2018.2/94095> , hdi:10013/epic.da16/8ed 62c3 4015 ae89 ba3a7f6723a6, 2018.

Purser, A., Hehemann, L., Dreutter, S., Dorschel, B. and Nordhausen, A.: OFOBS Seafloor images from the Antarctic Peninsula and Powell Basin, collected during RV POLARSTERN cruise PS118, Alfred Wegener Inst. Helmholtz Cent. Polar Mar. Res. Bremerhav., doi:10.1594/PANGAEA.911904, 2020.

380 Rott, H., Skvarea, P. and Nagler, T.: Rapid Collapse of Northern Larsen Ice Shelf, Antarctica, Science, 271(5250), 788–792, doi:10.1126/science.271.5250.788, 1996.

Scambos, T., Hulbe, C. and Fahnestock, M.: Climate Induced Ice Shelf Disintegration in the Antarctic Peninsula, in Antarctic Peninsula Climate Variability: Historical and Paleoenvironmental Perspectives, pp. 79–92, American Geophysical Union (AGU). [online] Available from: https://agupubs.onlinelibrary.wiley.com/doi/abs/10.1029/AR079p0079, 2013.

385 Schoening, T., Purser, A., Langenkämper, D., Suck, I., Taylor, J., Cuvelier, D., Lins, L., Simon Lledó, E., Marcon, Y., Jones, D. O. B., Nattkemper, T., Köser, K., Zurowietz, M., Greinert, J. and Gomes Pereira, J.: Megafauna community assessment of polymetallic nodule fields with cameras: platform and methodology comparison, Biogeosciences, 17(12), 3115–3133, doi:https://doi.org/10.5194/bg/17-3115-2020, 2020.

Schröder, M., Ryan, S. and Wisotzki, A.: Physical oceanography during POLARSTERN cruise PS96 (ANT XXXI/2 390 FROSN), Alfred Wegener Inst. Helmholtz Cent. Polar Mar. Res. Bremerhay., doi:10.1594/PANGAEA.859040, 2016.

Shepherd, A., Wingham, D., Payne, T. and Skvarca, P.: Larsen Ice Shelf Has Progressively Thinned, Science, 302(5646), 856–859, doi:10.1126/science.1089768, 2003.

Skvarca, P.: Fast recession of the northern Larsen Ice Shelf monitored by space images, Ann. Glaciol., 17, 317–321, doi:10.3189/S0260305500013033, 1993.

FIGURES

Figure 1: The Ocean Floor Observation and Bathymetry System (OFOBS) of the Alfred Wegener Institute (AWI),
 Helmholtz Centre for Polar and Marine Research Bremerhaven, deployed from the RV *Polarstern* during cruise PS118 in the waters east of the Antarctic Peninsula and on the flanks of the Powell Basin (Feb – April 2019)